# Body, Scale, and Space: Study on the Spatial Construction of Mogao Cave 254

Weiqiao Wang [1] and Aibin Yan [2,*]

1   College of Architecture and Urban Planning, Tongji University, Shanghai 200092, China; wangweiqiao@tongji.edu.cn
2   Department of Landscape Planning and Design, School of Art Design and Media, East China University of Science and Technology, Shanghai 200237, China
*   Correspondence: yanaibin@ecust.edu.cn

**Abstract:** This article focuses on the relationship between body, scale, and space, as revealed in Mogao Cave 254 in Gansu Province. Three topics, namely, body scale, pilgrim behavior, and time–space perception, are discussed. A space model based on mapping and measurement by former scholars is created to facilitate and visualize the analysis of the body scale of the cave space; the restriction of body scale suggests certain pilgrim behavior in the cave, whereas the occurrence of body behavior results in perception in the dimension of time. How time and space are related must be understood to comprehend the motif of Buddhist expression. This study is an architectural approach to spatial analysis that integrates the design, construction, and use phases through the scale, behavior, and perception dimensions. It is dedicated to broadening and enriching the cognitive dimensions of the space value of Mogao caves to reveal the original value of caves as religious spaces and completely preserve their material and invisible cultural heritage.

**Keywords:** Mogao Cave 254; space; body scale; pilgrim behavior; time–space perception

## 1. Introduction

The Dunhuang Mogao Grottoes, which are called *Mogaoku* in Chinese, are one of the most important man-made grottoes and world heritage sites. These sites are located in a small, long, and narrow oasis in northwestern Gansu Province in China, approximately 25 km southeast of Dunhuang City. The caves face east of Sanwei Mountain and stretch north and south for approximately 1600 m at the foot of the Mingsha Mountains. During the 4th–14th centuries, it witnessed changes spanning more than 10 dynasties. A total of 735 caves were built, 492 of which were decorated with paintings and sculptures. Caves have six fundamental forms: meditation cave, central-pillared cave, assembly hall, huge image cave, nirvana cave, and residential cave. In the Northern Wei (445–534), the cave style in Dunhuang is in a period of formation. The central-pillared cave is the main type, illustrating the transformation from the India Chaitya cave style to the Chinese style. Cave 254, which is situated near the center of the cliff, is the earliest central-pillared cave among 10 caves built during the Northern Wei.

After the Dunhuang Library Cave was discovered in 1900, it was first visited and investigated by Aurel Stein in 1907 (Stein 1980). Inspired by Stein, many scholars have devoted their time and lives to investigating, numbering, recording, depicting, analyzing, and protecting the grottoes in Dunhuang. Cave 254 has been extensively studied by former scholars and has been studied in various aspects: mapping the cave to present its original size (Shi 1996), discussing the historical background and its central-pillared style from the view of cave construction (Wang 2013), analyzing Buddha's life tales from the perspective of arts (Li 2000), and investigating on the Thousand Buddha and the meaning of statues from the background of Buddhist studies in the Northern Wei Dynasty (Ning and Hu 1986, pp. 22–36). Considering the whole study on the cave, Abe (1989) conducted

a thesis on Cave 254 from a global standpoint, "*a case study—an intensive, contextual exami­nation of a single case … an attempt to illuminate the intricate interplay of local circumstances in the decisions reached and ultimately to understand Cave 254 in terms of these choices*". Recently, Chen and Chen (2017) conducted detailed and vivid research on the paintings of Cave 254 (Chen and Chen 2017). Wu (2022) experimented with a new way to research and under­stand the art of Dunhuang, revealing that the concept of "space" is used as an entry point to treat the Mogao Caves as historical places and sites that can be physically approached and entered and touched with the eye. Furthermore, he summarized his approach to the "spa­tial analysis of caves" in five levels: (1) multiethnic and cross-cultural historical sites; (2) religious art as a whole; (3) internal spaces containing architecture, sculpture, and frescoes and ritual sites where religious events were held and historical memory was preserved; (4) spatial interaction of frescoes, sculpture, and architecture in caves; and (5) virtual space–time guided by pictorial space (Wu 2022).

Sculptures and murals in caves are important and express the historical, cultural, and religious situation of the time. However, this study emphasizes the role of space as the bearer of its narrative motifs, which deserves more attention. First, cave space provides a specific spatial context for sculptures and murals. The spatial layout and surrounding elements can enhance or reinforce the intended narrative, cultural symbolism, or religious significance of the artwork. Second, the arrangement of space helps realize religious ritual and cultural practices. By examining spatial organization and its relationship to narra­tive motifs, we can gain insights into the ritualistic and symbolic aspects of a particular culture or religious tradition. Third, space, together with sculptures and paintings, is in­tegrated as a whole and becomes an intrinsic part of the spatial design, serving functional and symbolic purposes. The spatial relationship between the artwork and the surrounding architecture can offer insights into the broader cultural and religious intentions behind the construction and use of the space.

To understand the religious language revealed by the spatial ontology and its original spatial value, the article uses the body scale as a medium. On the basis of the studies of former scholars, we try to understand the cave for its original function, i.e., a sacred space, and its relationship with the body. The space of the cave is composed of architecture, murals, and sculptures. The body could be comprehended from three aspects: craftsmen, pilgrims, and Buddhas. The relationship between body and space is categorized as body scale, pilgrim behavior, and time–space perception.

As discussed by Wu (2022) in the conclusion of *Spatial Dunhuang, approaching the Mo­gao caves*, Dunhuang is a comprehensive treasure trove that can be studied and explored from the perspectives of art history, religious history, archaeology, architectural history, etc. However, we also see Dunhuang as a man-made object that can be explored more from the space itself. In contrast to the "spatial analysis of caves" discussed by Wu Hong, this study is an architectural approach to spatial analysis that integrates the design, con­struction, and use phases through scale, behavior, and perception dimensions.

## 2. Body Scale and Cave Construction

Cave 254 is a large, almost rectangular excavation with a central pillar extending from floor to ceiling in the rear section. It occupies an area of approximately 64 square meters (9.5 m deep and 6.7 m wide). It shares a similar layout with India Chaitya in that the central pillar is the symbol of a stupa, whereas the surrounding space is intended for pilgrims to admire and revolve around the Buddha. The ceiling is divided into a flat ceiling in the rear around the central pillar and a gabled ceiling in the front, which is influenced by traditional Han-style architecture. Except for the floor, all the surfaces of the cave are covered by sculptures and paintings.

As a Buddhist cave, the body scale plays an important role in the relationship between Buddha and the pilgrims. In particular, the dimension of Buddha statues and murals, as well as their positions, may deeply affect viewers' perspectives, which ultimately work on their perception of the motif of Buddhism. When pilgrims enter Cave 254, they feel like

they are entering the world of Buddhas. The forms of caves contain meaning and symbols of heaven and earth, which further express the relationship between pilgrims and Buddhas. Sculptures of Buddhas and Buddhist murals are the main visual subjects for pilgrims, and their forms, scales, and positions are fundamental when referring to their influence on pilgrims' perceptions.

Scholars have investigated the viewing angle of adoring Buddha and its effect on pilgrims. Fu (1988) noted that 15–30 degrees is the best viewing angle in the vertical direction when pilgrims look up at the face of Buddha sculptures. If the viewing angle exceeds 45 degrees, they will feel the sense of the sublime from the sculptures of Buddhas. If it exceeds 60 degrees, they may feel majestic (Fu 1988). On the basis of Mogao Cave 172, Wu (1992) analyzed how the paintings on both murals exemplified a pictorial formula of Buddhas during the Tang Dynasty and how the composition of the painting required a particular perspective from the viewer outside the picture, leading the viewer from rough seeing (cujian) to the "mind's eye" (xinjian), i.e., the creation of religious images (Wu 1992). In Cave 254, even though sculptures of Buddhas share a similar size with pilgrims, their position in high places easily makes pilgrims look up and pay respect, as depicted by Hu and Hu (2005, p. 53). Fu (2009) analyzed the importance of an angle of 30 degrees and its high frequency of being applied in the organization of architectural interior space and sculptures of Buddhas. In his article about the evolution of the layout of Buddhist architecture and the organization of sculptures of Buddhas inside the temples in early China (*Zhong Guo Zao Qi Fo Jiao Jian Zhu Bu Ju Yan Bian ji Dian Nei Xiang She de Bu Zhi* [中國早期佛教建築布局演變及殿內像設的布置]), he mentioned that pilgrims can adore the face of Buddhas with the view field within a 30-degree elevation angle from far to near. Based on the cases of Cave 4 in Maijishan built in the Northern Zhou Dynasty, the Nanchansi Temple, and the Eastern Hall of Foguang Si (Buddha Light Temple) built in the Tang Dynasty, the viewing angles from the door of the building to the top of the backlight of Buddha and from in front of the Buddha to the ushnisha on the head of Buddha are 30 degrees. This angle is not only good for comfortable viewing but also good for pilgrims concentrating on adoring Buddha without looking up and down (Fu 2009).

Regarding cave construction, scholars have performed much work on mapping caves. Xie (1955) recorded the whole dimension, as well as detailed data on all the sculptures inside the Dunhuang caves. In the book "*Dunhuang Yi Shu Xv Lu* (敦煌藝術叙錄)," he measured the length in Chi (尺), a traditional Chinese measuring unit used in the late Qing Dynasty (equal to 0.35 m). In terms of accuracy, he mentioned that there might be some mistakes because he performed the measurement within a short period, and he reorganized it 13 years later (Xie 1955, p. 1). Shi (1996) was the first to draw the plans, facades, and sections of the Mogao Caves in 1942. In his book "*Mogao Ku Xing* (莫高窟形) (Shi 1996)", the main data of the cave are presented. Later, a scaled detailed drawing of the plan and section (Peng 1982, p. 208) was completed by Sun Rujianof the Dunhuang Research Institute. It was published in the book "*Dunhuang Mogao Cave* (敦煌莫高窟)" in 1982. Additionally, Ning and Hu 1986, pp. 22–23) focused on the Thousand Buddha and measured mainly the inside facades (Ning and Hu 1986, pp. 22–43). Their measurements on Cave 254 are shown in Table 1 below:

(1) Measured by Xie Zhiliu in 1942 and was published in his book "*Dunhuang Yi Shu Xv Lu* (敦煌藝術叙錄)" in 1955.

(2) Measured by Shi Zhangru in 1942 and was published in his book "*Mogao Ku Xing* (莫高窟形)" in 1955.

(3) Measured by Sun Rujian and was published in the book "*Dunhuang Mogao Cave* (敦煌莫高窟)" in 1982.

(4) Measured by Ning Qiang and Hu Tongqing and was published in the article "*Dunhuang Mogao Ku Di 254 Ku Qian Fo Hua Yan Jiu* (敦煌莫高窟第254窟千佛画研究)" in 1986.

**Table 1.** Measurements on Cave 254.

| | 1 | | 2 | | 3 | | 4 | |
|---|---|---|---|---|---|---|---|---|
| | Length (m) | Max. Height (m) | Length (m) | Max. Height (m) | Length (m) | Max. Height (m) | Length (m) | Max. Height (m) |
| East wall | 9.87 | 4.36 | 6.8 | 4.0 | 6.89 | 4.00 | 6.90 | 4.00 |
| South wall | 10.08 | 4.36 | 9.5 | 4.7 | 9.51 | 4.68 | 9.80 | 5.00 |
| West wall | 9.87 | 4.36 | 6.7 | 4.1 | 6.62 | 4.10 | 6.65 | 4.10 |
| North wall | 10.08 | 4.36 | 9.6 | 4.7 | 9.58 | 4.70 | 9.50 | 5.00 |

Notes: The accuracy of decimal measurements can vary depending on the precision of the measurement method and the tools used by each scholar. In addition, according to the answer of Sun Yihua, daughter of Sun Rujian, before the emergence of the 3D laser measurement tool, given that the caves are not custom made, each surveyor will have deviations due to the position they stand in and the direction they measure.

Table 1 reveals that Shi Zhangru and Sun Rujian's measurements are similar to each other and are more consistent with the main dimensions of the cave. By contrast, Xie Zhiliu, Ning Qiang, and Hu Tongqing highlighted the details of Buddha statues, murals, and the Thousand Buddha, and the data on them are more precise than their measurements on the main dimensions of the cave. Thus, the space model is constructed on the basis of the drawing by Sun Rujian, the main data by Shi Zhangru, and the detailed data of the Buddhas by Xie Zhiliu, as shown in Figures 1 and 2.

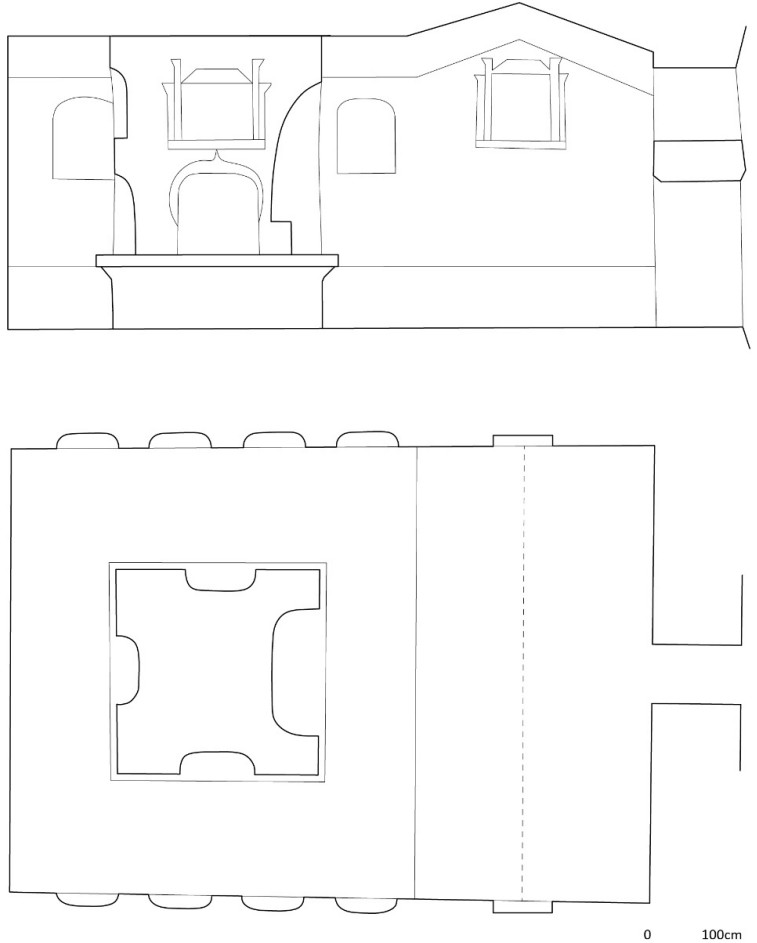

**Figure 1.** Plan and section of Cave 254.

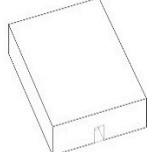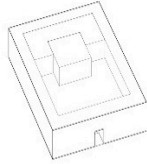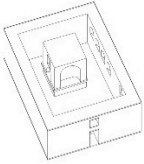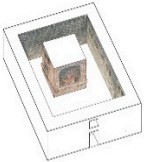

**Figure 2.** Construction process of Cave 254.

### 2.1. Craftsmen and Cave Construction

Craftsmen refer to the constructors of the Dunhuang Grottoes, as well as the first group of people who experienced the grotto space, and their importance cannot be ignored. According to the record on the construction process of the Dunhuang Grottoes (*Yin Ku Gao* [营窟稿]) (Ma 1997, p. 16) written at the beginning of the 10 century, recruiting excellent craftsmen is the second most important work next to choosing the best rock. The first thing to do for the success of cave construction is to provide a suitable workplace for craftsmen. On the one hand, the excavation scale should be large enough to meet the craftsmen's needs for workplaces. On the other hand, a certain correspondence exists between the size of the grotto and the body of the craftsmen.

The construction procedure is complicated for the following reasons: First, in contrast to the India Chaitya, which is composed of stone, engraving too much in Mogao Cave is not suitable because the cave is formed by the deposition of sand and pebble precipitation. Rock with looseness is better for painting than carving (Dunhuang Yan Jiu Yuan and Gansu Sheng Wen Wu Ju 2012). The minimal temperature changes in the caves ensure the excellent preservation of murals and sculptures. Therefore, according to Ma De, several kinds of craftsmen are needed during the delicate construction process: cave engravers, mud craftsmen (泥工), putty craftsmen (灰工), carpenters, sculptors, and painters (Ma 1997, pp. 17–22). Given the intricate procedure, the craftsmen group, which is a hugely complex system, is organized to enter the cave in order.

The construction process of Cave 254, as shown in Figure 2, can be summarized in four steps: (1) A gate hole 2.1 m high and 1.4 m wide was dug out so that two craftsmen and excavation tools could get through at the same time. (2) A gabled ceiling and a flat ceiling were made around the central pillar by digging upward. Digging the cave in a top-down approach is more convenient and safer for the craftsmen because scaffolds are not required (Sun and Sun 2003). The spacious front chamber, which is 6.8 m wide and 4.4 m deep, allows more craftsmen to work simultaneously. The corridor around the central pillar is 1.57–1.7 m wide, thus enabling two craftsmen to dig together and two painters or carpenters to work back to back simultaneously. (3) After digging out the whole cave space with the scaffolds, craftsmen who are responsible for wall trimming and mud touching must smooth the surfaces of the cave. (4) Painters and sculptors enter the cave and visualize the Buddhist motif. During all these procedures, the body of the craftsmen is an efficient tool for examining the spatial scale of the cave.

### 2.2. Buddha Scale

Sculptures of Buddhas and Buddhist murals are protagonists of the space, where scale is important for the perception of pilgrims. Given that sculptures are three-dimensional objects, their size and interrelationship may affect the distance perceived by pilgrims. To a great extent, pilgrims' perception of Buddha is affected by its volume. For murals, the distance between the pilgrims and the image is blurry. The scenes, scales, and drawing techniques may either exclude the pilgrims from the paintings or include them so that they feel like they are inside the painting, which has been pointed out as pictorial space (Wu 2022). Therefore, research on the scale of sculptures and paintings of Buddhas can lay a solid foundation for the analysis of pilgrim behavior in the next section.

In Cave 254, more than 1300 Buddhas are expressed through sculptures and murals. A total of 20 Buddhas are presented on a large scale: seven Buddha statues on the central pillar, five on the north and south walls, and three in the murals. They can be understood as

three kinds of Buddhas: cross-legged Maitreya bodhisattva, dhyana Buddha, and preaching Buddha. All four walls, as well as the four sides of the central pillar, are divided into three registers. The upper register is about Apsaras, bodhisattvas, figurines, and heavenly musicians. The middle register focuses on Maitreya bodhisattva, dhyana Buddha, preaching Buddha, and the Thousand Buddha. The lower register is only about yakshas. Table 2 explains their layout, as well as the original height of each Buddha, based on the measurement by Xie 1955, pp. 326–29) in 1942.

With regard to the data shown above, several principles on the scale can be drawn from the layout of Buddha and the measurements of height. (1) Space itself is completely center-symmetrical; thus, Buddha statues corresponding to the north and south walls share the same scale. (2) The scale of the Buddha statues in the central pillar is larger than that of the Buddhas on the walls in the corresponding position. (3) The space in Cave 254 is separated into two chambers: the front chamber is mainly about Maitreya worship, whereas the central-pillared space focuses on preaching and meditation. Thus, the scale of Buddha statues, to a large extent, is affected by the motif. Maitreya bodhisattva is relatively larger than the preaching Buddha and dhyana Buddha. (4) Although it is the Buddha that is shaped and visualized, we find that the construction of Buddha statues follows the scale of the human body.

A few documents on the instructions for molding the figures of Buddha have been published. Existing documents, such as "*Zao Xiang Liang Du Jing* (造像量度經)," (Gongbuchabu 1874) were retranslated into Chinese from the Tibetan edition by Gongbuchabu in 1742 during the Qing Dynasty. Wang (2002) also discussed sculptures of Buddhas in the Qing Dynasty in his book "*Qing Dai Jiang Zuo Ze Li Hui Bian: Fo Zuo, Men Shen Zuo* (清代匠作則例彙編：佛作，門神作) (Wang 2002)". In reference to research on the shape and proportion of statues in the Northern Wei Dynasty, the article "*Bei Wei Luo Yang de Fo Jiao Shi Ku yu Yong Ning Si Zao Xiang* (北魏洛陽的佛教石窟與永寧寺造像)" written by Qian (2006) can be regarded as a more representative study. Based on Buddha statues in grottoes and the Yongning Temple, a more detailed study of their shape characteristics is conducted. In the Northern Wei Dynasty, for a sitting Buddha, the ratio of the height of its head to body is 1:4, and for a standing Buddha, the ratio is 1:6. The bodhisattva statue is slightly different. For the standing bodhisattva, the ratio is 1:5.5 or 1:6 (Qian 2006). These data basically match the proportion of the grotto statues in Dunhuang during the Northern Dynasties. According to the Archaeological Report of Cave 266–275 in Mogao Grottoes (*Mo Gao Ku Di 266–275 Ku Kao Gu Bao Gao* [莫高窟第266–275窟考古報告]), which collects the most accurate data on the Buddha statues in Mogao grottoes during the era of the northern dynasties, for standing Buddha, the ratio of the height of its head to body is 1:6, whereas for sitting Buddha, it is 1:4. For standing bodhisattva, the ratio is 1:5.5, whereas, for sitting bodhisattva, it is 1:3 to 1:3.5 (Dunhuang Yan Jiu Yuan 2011). The application of body proportion contributes to achieving the integrity of Buddha's space. Table 3 explains how the body scale is used in shaping two chambers, as well as the relationship between Buddha statues and pilgrims when the height of Buddha is converted to standing height.

**Table 2.** Figures on each surface of Cave 254.

| The figures on the central pillar | | | | | | |
|---|---|---|---|---|---|---|
| | East | | South | West | North | |
| Upper register | Apsaras and bodhisattvas | | Figurines | Figurines | Figurines | |
| Middle register | A cross-legged Maitreya Buddha (2.03 m) | | A cross-legged bodhisattva (0.91 m) | A dhyana Buddha (0.875 m) | A cross-ankled bodhisattva (0.91 m) | |
| | | | A dhyana Buddha (1.085 m) | A dhyana Buddha (1.19 m) | A dhyana Buddha (1.085 m) | |
| Lower register | Ten yakshas | | Eight yakshas | Six yakshas | Six yakshas | |
| **The figures on East wall** | | | | | | |
| Upper register | heavenly musicians | | | | | |
| Middle register | Thousand Buddha motifs | | | window | Thousand Buddha motifs | |
| Lower register | Guardian warriors | | | door | Guardian warriors | |
| **The figures on West wall** | | | | | | |
| Upper register | Eighteen heavenly apsaras | | | | | |
| Middle register | Thousand Buddha motifs | | | | | |
| | Thousand Buddha motifs | | A white-robed Buddha in Dharmachakra mudra (1.12 m) | | Thousand Buddha motifs | |
| Lower register | Seventeen yakshas | | | | | |
| **The figures on South wall** | | | | | | |
| | Under the gabled ceiling | | | Under the flat ceiling | | |
| Upper register | Nine heavenly musicians | | | Seventeen heavenly musicians | | |
| Middle register | Thousand Buddha motifs | A cross-legged bodhisattva (1.33 m) | Thousand Buddha motifs | A preaching Buddha (0.805 m) | A dhyana Buddha (0.805 m) | A preaching Buddha (0.805 m) | A dhyana Buddha (0.805 m) |
| | Illustration of Defeating Mara/Sakyamuni sits under the bodhi tree with his hands in Bhumyakramana mudra (0.77 m) | | The tale of the Sattva Jataka | Thousand Buddha motifs | A preaching Buddha | Thousand Buddha motifs | |

**Table 2.** *Cont.*

| Lower register | Twenty-five yakshas | | | | | | |
|---|---|---|---|---|---|---|---|
| The figures on North wall | | | | | | | |
| | Under the gabled ceiling | | | Under the flat ceiling | | | |
| Upper register | Nine heavenly musicians | | | Eighteen heavenly musicians | | | |
| Middle register | Two apsaras | A seated Buddha | Two apsaras | A preaching Buddha (0.805 m) | A dhyana Buddha (0.805 m) | A preaching Buddha (0.805 m) | A dhyana Buddha (0.805 m) |
| | Thousand Buddha motifs | A cross-legged bodhisattva (1.33 m) | Thousand Buddha motifs | | | | |
| | The karma story of Nanda/Sakyamuni sits preaching (0.77 m) | | The Sibi Jataka tale | Thousand Buddha motifs | A preaching scene of the white-robed Buddha | Thousand Buddha motifs | |
| Lower register | Twenty-five yakshas | | | | | | |

**Table 3.** Dimension of height of Buddha statues in Cave 254.

| | | Under Gabled Ceiling | | | Under Flat Ceiling | | | | |
| | | South Wall | Central Pillar | North Wall | South Wall | Central Pillar | | | North Wall |
| | | | | | | South | West | North | |
| Original height (m) | up | 1.33 | | 1.33 | 0.805 | 0.91 | 0.875 | 0.91 | 0.805 |
| | down | | 2.03 | | | 1.085 | 1.19 | 1.085 | |
| Converted to standing height(m) | up | 2.00 | | 2.00 | 1.21 | 1.67 | 1.31 | 1.67 | 1.21 |
| | down | | 3.19 | | | 1.63 | 1.79 | 1.63 | |

As shown in Table 3, the largest statue in Cave 254 is situated in the central pillar, where the cross-legged bodhisattva with a standing height of 3.19 m is on the east side and the dhyana Buddhas with a standing height of 2.00 m are on the south and north sides of the central pillar. Then, a dhyana Buddha on the west side, which is 1.79 m in standing height, ranks third. Even though their cross-legged and meditating posture would reduce their height, the lower register, which is 0.9 m in height, ensures that the Buddha statues still look higher than the pilgrims. The height of the rest of the Buddha statues ranges from 1.21 m to 1.67 m, which is similar to the average height of the pilgrims (1.674 m). However, being situated in niches, which are approximately 2.4 m from the ground, makes them look much taller.

The application of the scale of the Buddha statues is conducive to the formation of the two Buddha spaces: Maitreya bodhisattva space and meditating space. Seemingly, Maitreya was held at a higher level of importance than Sakyamuni during the Northern Wei Dynasty. Thus, the integrity of the space leads pilgrims to comprehend the Buddhist motif well.

*2.3. Pilgrim Scale*

The pilgrim scale is mainly affected by two aspects: the available space in the cave and the relative relationship between the pilgrim scale and the Buddha scale. In this regard, the reasonableness of the spatial scale has been tested with the craftsmen's bodies during the excavation of the cave. The real body scale perceived by pilgrims depends on the further decoration of the cave.

The front chamber under the gabled ceiling, which is 4.7 m high, 6.8 m wide, and 4.4 m deep, serves as a relatively spacious reception for pilgrims to stay and conduct pilgrim activities. The corridor around the central pillar is 1.57 m to 1.7 m wide, which is relatively narrow and high, allowing only one to two pilgrims to revolve around the Buddhas at the same time.

Additionally, the comfort of the body scale is affected by the scale of Buddha statues and the dimensions of murals. The discussion on the Buddha scale reveals that the dimension of the Buddha statues is larger than that of the pilgrim, and their position is higher than that of the pilgrims, making the pilgrims feel that Buddha statues are much larger than their actual size. In addition, murals are references for pilgrims to perceive their scale. The average height of the population during the Northern Wei Dynasty is 1.674 m (Zhang and Du 2008), which means that the position of their eyes is approximately 1.6 m above the ground. Usually, 20 and 36 degrees are comfortable viewing angles in the vertical and horizontal directions, respectively, ensuring a good visual presence without getting tired from the frequent rotation of the eyeball. The view field within 30 degrees of the view point, called the central view field, is good for observation (Zhao and Yang 2013). When the pilgrims stand in front of the central pillar, under the beam of the gabled ceiling, the dimension of the image on the wall that can be seen comfortably is 1.2 m high and 2.2 m wide, as shown in Figure 3. As shown in Table 4, the dimension of murals mostly satisfies the visual scale of pilgrims. In addition, given the below register of yakshas, the height of the center of the mural is approximately 1.67 m, which is the same as the average height of

the pilgrims' eye level. They could enjoy a better sight of it, including stepping forward to appreciate the details. In addition, the height of the eye level of Buddha in the illustration of Buddha's Vanquishing Maras and the Pravajana of Nanda is approximately 2 m, which is slightly higher than the pilgrim's eye level, making the pilgrims look up. Such a relationship on the scale is not a coincidence, and we can speculate that painters may decide the dimension of the murals with reference to their body scale and perception. By contrast, the narrow corridor around the central pillar is not expected to be comfortable for pilgrims to observe the whole mural. It leads pilgrims to focus more on their inner peace, revolving around the central pillar with a glance at the details of the Thousand Buddhas.

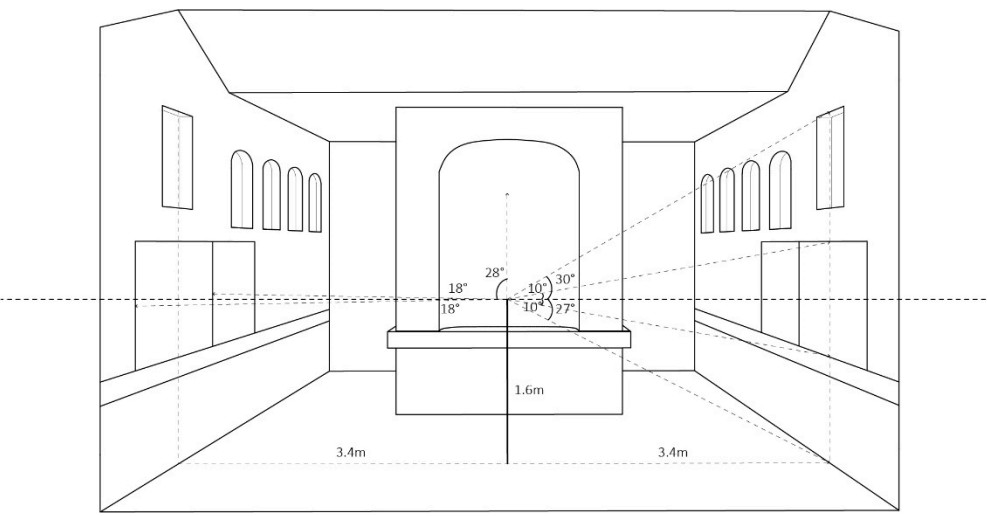

**Figure 3.** Analysis of the visual field in Cave 254.

**Table 4.** Dimension of murals (and the height of Buddha).

|  | South Wall | | North Wall | | West Wall |
|---|---|---|---|---|---|
|  | **Illustration of Buddha's Vanquishing Maras** | **Jataka of Mahāsattva with Seven Episodes** | **Jataka of Sivika with Five Episodes** | **Pravajana of Nanda** | **A Preaching Scene of a White-Robed BUDDHA** |
| Width (m) | 1.785 (0.875) | 2.7825 | 1.75 | 2.8 (0.875) | 1.19 |
| Height (m) | 1.54 (0.77) | 1.54 | 1.575 | 1.575 (0.77) | 1.12 |
| Distance to the Ground (m) | 0.9 | 0.9 | 0.9 | 0.9 | 0.9 |

## 3. Body Behavior and Worship Liturgy

*"As both created and creator, a religious building manifests the aspirations and intentions of its builders, yet the meaning of a building 'not occasionally, but always' surpasses those original intentions."* (Jones 1993)

Although the body scale of the craftsmen, Buddha statue, and pilgrim is the initial standard to measure the feasibility of the cave size, according to Jones (1993), the meaning of the space is far beyond the original intention of the creator, and the behavior and rituals evoke feelings and ideas that far exceed the spatial arrangement of the cave. By contrast, today's Mogao Cave has become a dead cultural relic, i.e., without rituals and pilgrims. In addition, talking about the possible religious rituals of the past dynasties, including the different changes in the rituals, is difficult. According to the analysis of Jinci, the sacred Chinese ritual architecture by Miller (2007), pious pilgrims will be lost without rituals, and the objects of worship may also change because of this.

*"Su You indicates with this inscription that the presence of the divine and the sincerity of the devotee were lost when the building in which the ritual took place was not main-*

*tained. Facing the loss of his residence and, as a consequence, the impossibility of patrons' performing a sincere ritual for invoking the spirit, Shu Yu would no longer descend and would be unable to help 'transform' the local people."* (Miller 2007)

Therefore, the following analysis of behavior and rituals is deduced more from the perspective of the spatial arrangement itself. However, as Jones (1993) said, in the dialog between the little boy and the statue, the details of such behavior and spatial relationship are beyond our imagination:

*"And so, while this meticulously choreographed mass with music, vestments, scriptural readings, and holy sacraments was being performed for hundreds of people in the congregation, this little boy spent the hour in the side aisle involved in a very animated conversation with this same-sized stone angel. He greeted her nose-to-nose, put his hands all over her, interrogated her, and then stepped back fully expectant, so it seemed, of a response."* (Jones 1993)

Even though ritual itself is not static and always the same, analysis of the *"interaction of architecture and ceremony in sacred places for its sacred value"* (Wescoat and Ousterhout 2014) is necessary. This article tends to start with elements that can be measured and judged at the spatial level. The body scale of Cave 254 ensures that certain behaviors occur in space. On the one hand, it meets the needs of the Northern Wei Dynasty, that is, Maitreya worship and meditation are mainstream in admiring Buddha; on the other hand, it encourages pilgrims to enter and understand the realm of Buddha through corresponding behavior. Therefore, the cave space has two main sections: the former space under the gabled ceiling and the central-pillared space under the flat ceiling. It provides space for monks and pilgrims to gather and enables them to revolve around the Buddha in the rear (Xiao 1989). Regarding subjects in different grottoes according to different periods, Duan et al. (1995) pointed out that *"In the early grottoes (during the Sixteen States* [304–439] *and the North Dynasty* [439–534]*), much expression was given to the attainment of Buddhahood through the practice of satparamita and cetana. The Dunhuang artists put special emphasis upon subjects concerning vipasyana, dhyana, and samadhi."* (Duan et al. 1995). Thus, the Buddhist motif expressed in Cave 254 and the way of organizing the space were influenced by mainstream beliefs. Finally, pilgrims' behaviors in caves are greatly affected by spatial scales, statues, and murals.

### 3.1. Worshiping the Maitreya Bodhisattva

The symmetry of the cave stresses the importance of the central area. When pilgrims enter the cave through the door, they first encounter a cross-legged Maitreya bodhisattva in the central pillar (Figure 4), which is lit from the window and the diffuse light from the floor. Naturally, pilgrims would look up and worship the bodhisattva.

Then, they may look around in the front chamber where cross-legged Maitreya bodhisattvas are set in the upper niches on the walls, as well as the central pillar, which is the symbol of living in the Tusita Temple (兜率天宫). According to the Maitreya sutra *Mi Le Xia Sheng Jing* (弥勒下生经), which was translated by *Kumarajiva* (鸠摩罗什), Maitreya Bodhisattva is the future Buddha, who is entrusted to the care of hope for the pilgrims. The Tusita Temple is extremely wonderful that if they could practice Buddhism devoutly and meditate, they could enter the Elysium afterlife (Peng 1982, p. 167). Therefore, Buddha's life stories and preaching and meditating scenes are shown on the murals, which are the best examples for the pilgrims.

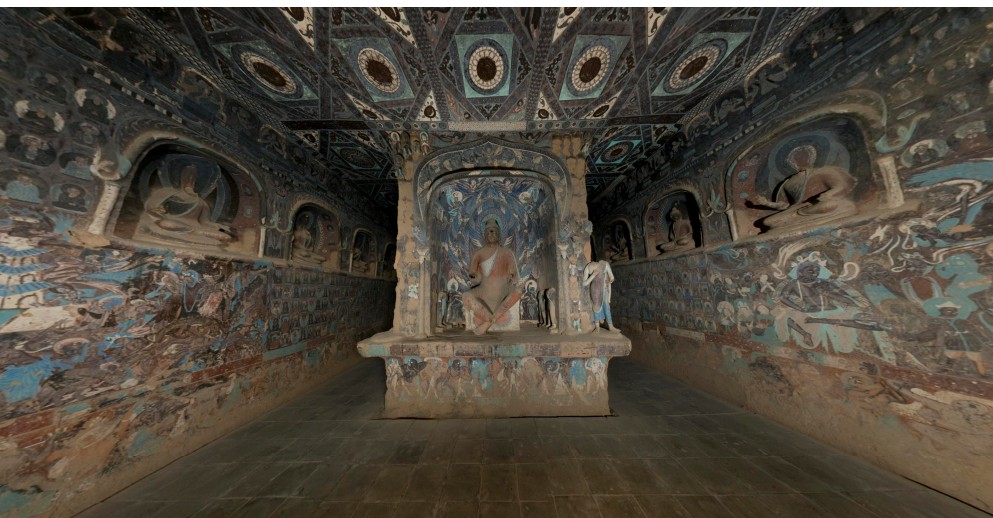

**Figure 4.** Entrance view of Cave 254, image cited from Digital Dunhuang, https://www.e-dunhuang.com/cave/10.0001/0001.0001.0254. Accessed on 17 July 2023.

*3.2. Watching Buddha's Life Stories*

After worshiping the Bodhisattva, pilgrims may stand in the center under the gabled ceiling space, where pilgrims feel comfortable appreciating the whole murals on both walls (Figure 5). The Buddhas' virtue is illustrated vividly by the mural for the pilgrims. Under the gabled ceiling, Jataka tales and Buddha's life stories are illustrated on the wall, whereas the dimension gives enough space for watching and observing the drawings. After worshiping the Buddha, pilgrims can turn around to appreciate the murals. The contents of the murals are full of dynamic sections. For example, Mahasattva Jataka on the south wall illustrates Prince Sattva offering himself to a starving tigress and her cubs. The scene consists of seven episodes within a single rectangular space: finding the tigress, piercing his neck, jumping from the cliff, feeding the tigress, remains being found by the family, crying on the remains, and building the stupa. This drawing has had many interpretations, but the pilgrims watch the dynamic stories of the mural in a still manner.

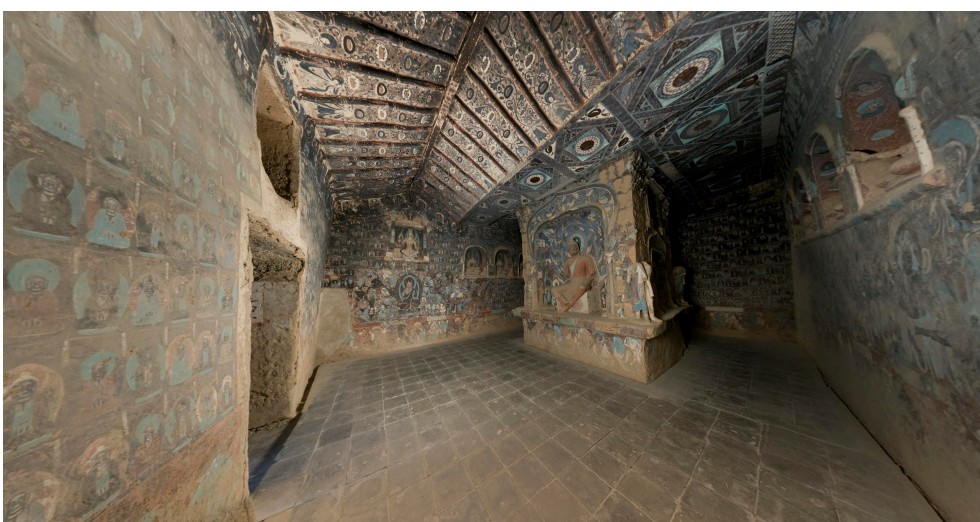

**Figure 5.** View of the central pillar from the corner under the gable ceiling space of Cave 254, image cited from Digital Dunhuang, https://www.e-dunhuang.com/cave/10.0001/0001.0001.0254.2.3. Accessed on 17 July 2023.

The space around the central pillar is dominated by revolving around the Buddha. Four Buddha statues are carved in the pillar, and two cross-legged bodhisattvas are carved

on the north and south surfaces. To obtain a whole view of all the Buddhas, pilgrims must inevitably walk around the pillar. The narrow corridor, which is only 1.57 m wide, is not comfortable enough for pilgrims to view the murals on the wall. Therefore, the scenes of hell at the bottom and heaven at the top are out of a suitable viewing angle, which is perceived from the pilgrims' peripheral vision. The central-pillared space is mainly about preaching and meditation. The four walls are adorned by the Thousand Buddha motif: the Buddhas of the past, present, and future ages, which together form a great scene of preaching and meditation. This space, even without pilgrims, is complete. The Buddhas, who have six supernormal cognitive abilities, could sense limitlessness in a second, which is not restricted by the distance and the viewing angle. Pilgrims simply revolve around the Buddhas, and even though the murals of the Thousand Buddhas and the sculptures are still, pilgrims admire the still images in a dynamic way. It finally transforms the still image into a dynamic and continuous horizontal scroll, on which time and space have a magic influence, expressing the time–space value of the reincarnation of Buddhism.

## 4. Perception, Time, and Space

*"Like a condensed journey through time and space, the vast expanse of time and space is arranged in an orderly manner in a cave of less than 65 square metres, and the Buddhist cosmology and worldview they embody is still clear and thought-provoking to today's viewers."* (Chen and Chen 2017, p. 81.)

The construction of the cave is a kind of vivid visualization of the world of Buddhas, which intends to raise resonance from the inner spiritual space of the pilgrims. The intention of becoming Buddha is not something newly created. However, it is the root that embeds in each one's mind, which they used to have and now forget. To some extent, the perception of space serves as the foundation for the pilgrims to cover thoughts on time, which finally opens the door to the realm of Buddhas for them.

### 4.1. Space Perception

Defining space itself is difficult, and we normally rely on reference points to depict the limit of the space. Architectures, murals, and sculptures are tangible references to space. The relationship between the reference and the body could be sensibly understood as perception while being rationally understood as scale. Usually, perception and behavior result in certain behavior in the space. The perception of comfort depends on the relationship between the dimension of space and the human body. Spacious and bright chamber spaces make people feel relatively relaxed, whereas the dim and narrow corridor around the central pillar leads pilgrims to focus more on their inner peace.

Regarding Cave 254, the six inside surfaces are vital references for pilgrims to perceive the border of the space. Most of the murals, sculptures, and construction details are expressed in the four facades, as well as the ceiling, to draw more attention upward. Simply looking up is not enough to understand the Buddha space fully. To understand it fully, pilgrims must observe the murals in the correct order, beginning with the depiction of hell, then to Buddha's life story, on to a flying apsaras in heaven, and ending on the ceiling with a symbol of reincarnation. The wall is a powerful medium for gaining space perception. Accompanied by murals and sculptures, a rigorous and orderly level of spatial perception could be created. With the process of looking up, pilgrims could fully comprehend the content expressed by the layers of the picture ascendingly.

Light is an intangible reference to space. Entities, such as walls, are objective factors making up space, whereas light is the determinant of the degree to which space is perceived. The bright and open lobby space contrasts with the dim and suppressed central-pillared space, which adds to the mystique of the cave motif. With the original trapezoidal perspective of the space, the space behind the center pillar seems to disappear.

Although walls could be the strong border for perception, light plays an important role in dominating pilgrims' perceptions and subjective emotions. The former space of Cave 254 is lit up by the window and door in the east wall where the Maitreya Buddha

is sacred and bright for pilgrims to admire. In addition, the bright and spacious chamber under the gabled ceiling allows pilgrims to stop in front of the grottoes and observe the details of the depiction. By contrast, the gloomy space around the central pillar gives one a feeling of mystery, which is good for meditation.

*4.2. Time Perception*

How to express unlimited time based on limited space is a challenging theme. Time itself is intangible, whereas space is expected to serve as a reference for pilgrims to perceive the track of time. In addition, behavior is an effective catalyst for participating in the operation of time, and pilgrims eventually obtain the perception of time in practice.

Space is the reference for time. The murals and sculptural expressions at Cave 254 provide pilgrims with the possibility of a multidirectional perception of time. In the front chamber, pilgrims can perceive the temporal and spatial changes in the picture. Worshiping the Buddhas and viewing the murals are similar to observing static images in a still manner. In the central-pillared space, pilgrims revolving around the Buddhas according to the central pillar is a kind of dynamic behavior compared to the still Thousand Buddhas and sculptures of Buddhas in the niches.

Buddhism's perception of space is not measured by material reference but by the description of time to show the understanding of space, such as the six realms of existence and reincarnation. In Cave 254, the space from the gabled ceiling to the central pillared forms a fixed sequence for pilgrims to go to the core space through instruction space. It is the space for Buddhas to meditate but not for pilgrims. Thus, it is impressive when pilgrims enter the mind palace of Buddhas.

Behavior is the catalyst for awareness. How can an unlimited timeline be perceived through limited space? The moving body is the carrier of perception. Usually, the dimension of time is realized through the motion of a body in space (Lou 1999, p. 165). When pilgrims revolve around the central pillar, the 1235 Buddhas on the four inside walls perform as a time belt, where the Buddhas of the past and future ages are set in clockwise order (Ning and Hu 1986, p. 30), leading pilgrims to walk around the central pillar clockwise to experience the time scale and to understand that the past, the present, and the future are a cycle.

According to Chen and Chen (2017), "the designers of Cave 54 have made all the Buddhas appear simultaneously, and as the faithful circumambulate the central pillar, they witness the thousand Buddhas of the past, present and future one by one, and people bound to a finite life receive the eternal blessing of the entire universe." (Chen and Chen 2017, p. 203)

In the course of the continuous circuit of Buddhas, the cave is dotted with images of Buddhas and beings as if they were reincarnated into the world from the past, present, and future, with the larger-scale images being the focal point of vision. By contrast, mural paintings, such as Prince Sattva Jataka, Shibi Jataka, Story of Nanda, and Vanquishing Mara, draw us into the specific details of Buddhahood and practice, in what can be described as a journey through time and space to the broadest and to the most subtle.

**5. Conclusions**

The grottoes' excavation and emphasis on meditation were important characteristics of Buddhist belief during the Northern Wei Dynasty (Tang 1997). Buddhist grotto arts went through from no image to a great image of Buddha; it not only satisfied the requirements of the public but also became a compulsory course on Vipassana (meditation) for monks. The early caves mainly focused on viewing sculptures of Buddhas. From the perspective of the existing grotto arts, viewing sculptures of Buddhas inside the caves and the Maitreya beliefs were even more prosperous during the Northern Wei Dynasty. Therefore, the size, scale, position, and style of the Buddha image are crucial to the pilgrims' perception. The spatial construction of Mogao Cave 254 depicts an all-encompassing world of Buddhism, with more than 1300 images and sculptures of Buddhas on different scales. It illustrates

well the relationship between body and space. Through this case study, we learn that the interrelationship between humans' and Buddha's scale is fundamental in affecting the pilgrims' perception of the Buddhist motif. In addition, architecture, murals, and sculptures play important roles in spontaneous pilgrim behavior, leading them to understand the paradise of Buddhas and the way of achieving inner peace. As Jones (1993) said, *"Religious buildings arise as human creations, but they persist as life-altering environments; they are, at once, expressions and sources of religious experience."*

Regarding Cave 254 as an overall architectural space from the perspective of body scale and spatial relationship is good for understanding the outstanding universal value of the entire historical scene. Based on the analysis of detailed mapping data, this article tries to reveal the construction purpose and restore a real construction scene of architectural heritage. Understanding the Buddha's life stories and observing Buddha sculptures help us learn vivid historic, artistic, and religious scenes. However, to comprehend fully the space of Cave 254 and its meaning in the Northern Wei Dynasty, one must step into the cave. Pilgrims use their body scale to measure the size of space, revolving around the central pillar to worship the Buddha and immerse themselves in the Buddhist atmosphere.

*"The ultimate goal of conservation as a whole is not to conserve the paper, but to retain or improve the meaning it has for people."* (Muñoz Viñas 2005)

When it comes to protecting the Mogao Caves, it is crucial to implement measures that effectively address major threats such as unstable rock structures, wind and sand hazards, water-related issues, and tourism pressures. The primary objective is to ensure that the caves are preserved with minimal alterations. The preservation of the Mogao Caves extends beyond safeguarding a physical site; it aims to preserve and present a relatively intact spiritual space for both current and future generations. In addition to mitigating the impact of visitation on the site's conservation, it is essential to convey the authentic and complete significance of the grottoes to visitors in a more specific manner. Presently, the Mogao Caves are open to the public, allowing access to certain caves on designated days. However, can the experience of the Mogao Caves be enhanced by adhering to specific behavioral rituals as it was in the past? By doing so, the human-created space of the grottoes could enable visitors to deeply immerse themselves in the profound atmosphere, even in this new era of digital heritage presentation.

Currently, a significant focus of the Dunhuang Academy's work is on digitization. The digitization project initiated by the Dunhuang Research Academy in 2006 serves not only long-term conservation and research purposes but also helps alleviate the conflict between tourism demands and heritage protection, to some extent. Ideally, through digital presentation, viewers can fully grasp the profound significance of the caves. However, the Digital Dunhuang project primarily provides a basic representation of the grotto spaces, presenting the interior situations of the caves in a uniform manner. Based on the analysis, it is possible to further enhance the presentation of specific caves, such as Cave 254 in the Mogao Caves, by offering a distinct perspective that allows visitors to comprehend the cave from multiple dimensions. Understanding the overall spatial scale of Cave 254 is crucial in recognizing the complete artistic value of its engineering-oriented practice in the spatial context. The methodology employed in this study can be applied to investigate caves from different time periods, thus aiding in understanding the evolving relationship between pilgrims and Buddhas over time. Visitors are guided to experience the grotto from a concrete viewpoint, even if it is through the use of 3D visual effects, providing a simulated visitation experience.

**Author Contributions:** Conceptualization, W.W. and A.Y.; methodology, W.W. and A.Y.; formal analysis, W.W. and A.Y.; investigation, W.W. and A.Y.; resources, W.W. and A.Y.; writing—original draft preparation, W.W.; writing—review and editing, W.W. and A.Y.; visualization, W.W.; supervision, A.Y.; project administration, A.Y.; funding acquisition, A.Y. All authors have read and agreed to the published version of the manuscript.

**Funding:** The research was funded by Shanghai Pujiang Program, grant number 2020PJC021.

**Institutional Review Board Statement:** Not applicable.

**Informed Consent Statement:** Not applicable.

**Data Availability Statement:** Not applicable.

**Acknowledgments:** This article is conducted with financial support from the Shanghai Pujiang Program, grant number 2020PJC021. The draft of the article was written in 2018 when we were Visiting Fellows and participated in the course "Buddhist Monuments of the World" taught by Yukio Lippit, Eugene Wang and Jinah Kim at Harvard University. We are very grateful to them for their valuable suggestion and inspiration in this topic. Additionally, we would like to express our appreciation to Yingchun Li for providing constructive comments on the article.

**Conflicts of Interest:** The authors declare no conflict of interest.

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
