# Peer review of "Body, Scale, and Space: Study on the Spatial Construction of Mogao Cave 254"

_religions, doi:10.3390/rel14070953_

Round 1
Reviewer 1 Report
The topic is interesting, and the perspective chosen for entering into the subject-matter is excellent. There are several interesting remarks along the text. The argument is not always adequately expressed. Fr what I have understood this study differs form previous ones by (a) taking the scale of the human body as a hallmark of space analysis; (b) focusing also (though this part is not very developed) on the ritual use of the space visited by the pilgrims.
Some difficulties are linked to language issues. See for instance sentences such as:
- "There is no doubt that narrative motifs like sculpture and painting are much easier to draw attention, which leads to further understanding of the history, culture and religious situation of that time. " (The overall meaning is clear by the reader's attention is distracted by the awkward formulation.)
- "As what Wu (2022) has discussed" (ibid.)
- "Craftsmen were the first to experience the 149 grotto space, whose importance could not be ignored" The experience of the craftsmen?
- " (The accuracy of decimal depends on each one's measurement)" Meaning unclear
etc.
Although I chose "moderate English editing" because the syntax is generally correct, one may speak of "extensive Chinese editing" when it comes to the specification of the meaning.
The difficulty is increased by a certain imprecision as to the specificity of the contribution when compared to former studies. One "guesses" what is new in this article without being sure one can "ascertain" it.
Finally, the part on space and ritual is underdeveloped and lacks theoretical references (as well as comparative perspective). I would recommend that the authors refer to:
Jones, Lindsay. 1993. The Hermeneutics of Sacred Architecture: A Reassessment of the Similitude between Tula, Hidalgo and Chichem Itza, Yucatan, Part I. History of Religions 32 (2 parts)
Jones, Lindsay. 2000. The Hermeneutics of Sacred Architecture: Experience, Interpretation, Comparison, Cambridge: Harvard University Center for the Study of World Religions.University (2 vols)
Miller, Tracy. 2007. The Divine Nature of Power, Chinese Ritual Architecture at the Sacred Site of Jinci. Cambridge and London: Harvard University Press.
Wescoat, Bonna D., and Robert G. Ousterhout. 2014. Architecture of the Sacred: Space, Ritual, and Experience from Classical Greece to Byzantium. Cambridge: Cambridge University Press.
This is not anecdotical: being focused on the Dunhuang example the authors eventually lack the tools and perspectives that would help them to develop intuitions that seem to me basically sound.
Moderate editing needed when it comes to sntax
More extensive editing needed when its comes to specifying the purpose and method of this contribution
Author Response
The topic is interesting, and the perspective chosen for entering into the subject-matter is excellent. There are several interesting remarks along the text. The argument is not always adequately expressed. Fr what I have understood this study differs form previous ones by (a) taking the scale of the human body as a hallmark of space analysis; (b) focusing also (though this part is not very developed) on the ritual use of the space visited by the pilgrims.
Thanks for your comments, we have revised the article and developed the part on the ritual use of the space visited by the pilgrims. Please find them in the article.
Some difficulties are linked to language issues. See for instance sentences such as:
- "There is no doubt that narrative motifs like sculpture and painting are much easier to draw attention, which leads to further understanding of the history, culture and religious situation of that time. " (The overall meaning is clear by the reader's attention is distracted by the awkward formulation.)
Sculptures and murals in caves are important and express the historical, cultural, and religious situation of the time. However, this study emphasizes the role of space as the bearer of its narrative motifs, which deserves more attention. First, cave space provides a specific spatial context for sculptures and murals. The spatial layout and surrounding elements can enhance or reinforce the intended narrative, cultural symbolism, or religious significance of the artwork. Second, the arrangement of space helps realize religious ritual and cultural practices. By examining spatial organization and its relationship to narrative motifs, we can gain insights into the ritualistic and symbolic aspects of a particular culture or religious tradition. Third, space, together with sculptures and paintings, is integrated as a whole and becomes an intrinsic part of the spatial design, serving functional and symbolic purposes. The spatial relationship between the artwork and the surrounding architecture can offer insights into the broader cultural and religious intentions behind the construction and use of the space.
- "As what Wu (2022) has discussed" (ibid.)
As discussed by Wu (2022)
- "Craftsmen were the first to experience the 149 grotto space, whose importance could not be ignored" The experience of the craftsmen?
Craftsmen refer to the constructors of the Dunhuang Grottoes, as well as the first group of people who experienced the grotto space, and their importance cannot be ignored.
- " (The accuracy of decimal depends on each one's measurement)" Meaning unclear
etc.
The accuracy of decimal measurements can vary depending on the precision of the measurement method and the tools used by each scholar. In addition, according to the answer of Sun Yihua, daughter of Sun Rujian, before the emergence of the 3D laser measurement tool, given that the caves are not custom made, each surveyor will have deviations due to the position they stand in and the direction they measure.
Although I chose "moderate English editing" because the syntax is generally correct, one may speak of "extensive Chinese editing" when it comes to the specification of the meaning.
Thanks for pointing out the need to further improve the expression of the article. It has been proofread by professional English speakers. Please find it in the revised manuscript.
The difficulty is increased by a certain imprecision as to the specificity of the contribution when compared to former studies. One "guesses" what is new in this article without being sure one can "ascertain" it.
Regarding the innovation and contribution of the article, it has been added and strengthen in the part of conclusion.
Finally, the part on space and ritual is underdeveloped and lacks theoretical references (as well as comparative perspective). I would recommend that the authors refer to:
Jones, Lindsay. 1993. The Hermeneutics of Sacred Architecture: A Reassessment of the Similitude between Tula, Hidalgo and Chichem Itza, Yucatan, Part I. History of Religions 32 (2 parts)
Jones, Lindsay. 2000. The Hermeneutics of Sacred Architecture: Experience, Interpretation, Comparison, Cambridge: Harvard University Center for the Study of World Religions.University (2 vols)
Miller, Tracy. 2007. The Divine Nature of Power, Chinese Ritual Architecture at the Sacred Site of Jinci. Cambridge and London: Harvard University Press.
Wescoat, Bonna D., and Robert G. Ousterhout. 2014. Architecture of the Sacred: Space, Ritual, and Experience from Classical Greece to Byzantium. Cambridge: Cambridge University Press.
This is not anecdotical: being focused on the Dunhuang example the authors eventually lack the tools and perspectives that would help them to develop intuitions that seem to me basically sound.
Thanks a lot for your suggestions. We have carefully read these articles and books, and some of them are referred to in the manuscript.
Reviewer 2 Report
The Article is clean and well supported via data, citations, and cogent reasoning. I accept it in current form.
Author Response
Dear Reviewer,
Thanks for your comments and encouragement.
Best regards.
Reviewer 3 Report
It is hard to describe the originality of this research and what the author tries to argue and contribute to the scholarship. Neither the writing nor the tables are helpful for the readers to understand the main point. This topic should have been better and more heavily illustrated.
Extensive editing of the English language is required. In addition to grammatical errors, the organization of sentences and paragraphs should also be improved so the ideas can be better presented. Some words might be misused, such as ”map" or "mapping." The meaning of "scale" is not clear or consistent when used in different contexts.
Author Response
Thank you for your comments. The article has been revised, adjusted, and proofread in English, and images have been added to enhance clarity of expression. Please find detailed revisions in the manuscript.
Round 2
Reviewer 1 Report
The clarity of the article has been considerably improved.
i would have extended the analysis of the interaction between the space and rituals/religious practitioners, but, as it is, the paragraph dedicated to the issue gives indeed accrued depth to the analysis.
Improvements have been made, and, even if some sentences are a bit awkward, the reading is generally smooth.
A last rereading is needed. Some expressions are surprising and need rephrasing even if the meaning is clear ("The grottoes’ excavation and emphasis on meditation", "Even though ritual itself is not static and always the same as it was,", "engraving too much in Mogao Cave is 172 not suitable", "the viewing angle on adoring Buddha")
Reviewer 3 Report
This manuscript is significantly improved compared to the last version.